# The Role of Vitamin D in the Development of Diabetes Post Gestational Diabetes Mellitus: A Systematic Literature Review

**DOI:** 10.3390/nu12061733

**Published:** 2020-06-10

**Authors:** Amélie Keller, Carmen Varela Vazquez, Rojina Dangol, Peter Damm, Berit Lilienthal Heitmann, Mina Nicole Händel

**Affiliations:** 1Research Unit for Dietary Studies at The Parker Institute, Bispebjerg and Frederiksberg Hospital, Part of the Copenhagen University Hospital, 2000 Frederiksberg, Denmark; carmenvarela@ub.edu (C.V.V.); trustisfact@gmail.com (R.D.); Berit.Lilienthal.Heitmann@regionh.dk (B.L.H.); Mina.Nicole.Holmgaard.Handel@regionh.dk (M.N.H.); 2Department of Clinical Psychology and Psychobiology, University of Barcelona, 08035 Barcelona, Spain; 3Center for Pregnant Women with Diabetes, Department of Obstetrics, Rigshospitalet, 2100 Copenhagen, Denmark; nis.peter.damm@regionh.dk; 4Department of Clinical Medicine, Faculty of Health and Medical Sciences, University of Copenhagen, 2200 Copenhagen, Denmark; 5Department of Public Health, Section for General Practice, University of Copenhagen, 2014 Copenhagen, Denmark; 6The Boden Institute of Obesity, Nutrition, Exercise & Eating Disorders, Charles Perkins Centre, University of Sydney, Sydney, NSW 2006, Australia

**Keywords:** gestational diabetes, vitamin D, type 2 diabetes

## Abstract

Women diagnosed with gestational diabetes mellitus (GDM) are more likely to later develop diabetes. Evidence from some previous reviews suggests that low vitamin D status during pregnancy increases the risk of developing GDM, but whether vitamin D during pregnancy also influences the risk of diabetes post GDM is less well studied. Thus, the aim of this systematic literature review was to summarize the current available literature on that topic. This review considered observational studies and randomized controlled trials (RCTs). Five databases were searched. The risk of bias of the included studies was assessed. A total of six studies were included: three observational studies and three RCTs. Findings were inconsistent across the six included studies. However, when considering RCTs only, the findings more strongly suggested that vitamin D supplementation during and after pregnancy did not have an influence on markers of diabetes development or diabetes development post GDM. This systematic review highlights inconsistent findings on the associations between vitamin D supplementation or concentration during and after pregnancy and markers of diabetes development or diabetes development post GDM; and although results from randomized interventional studies more strongly suggested no associations, the conclusion holds a high degree of uncertainty.

## 1. Introduction

Vitamin D is both a fat-soluble vitamin and a secosteroid obtained either from the diet as D_2_-ergocalciferol from vegetables and D_3_-cholecalciferol from foodstuffs such as oily fish and dairy products, or from fortified foodstuffs and supplements. The major source of vitamin D_3_ is, however, its subcutaneous synthesis after exposure to sunlight [1]. Vitamin D from the diet or from sun exposure is biologically in an inactive form; hydroxylation in the liver, to 25-hydroxycholecalciferol (25(OH)D), and in the kidney, to 1,25-dihydroxycholecalciferol (1,25(OH)D), is required for its activation [2]. 25(OH)D is considered to be the best biomarker of vitamin D concentration in the human body [3].

Gestational diabetes mellitus (GDM) is defined as glucose intolerance with first onset or recognition during pregnancy and is typically diagnosed during the second or third trimester of pregnancy but is not overt diabetes mellitus [4]. The prevalence of GDM is increasing worldwide and it is estimated that one in seven pregnancies is affected by GDM with an estimated worldwide prevalence between 1% and 35% [5,6]. Women diagnosed with GDM are more likely to develop type 2 diabetes mellitus (T2DM) later on [7,8].

Evidence from observational studies suggests inverse associations between 25(OH)D concentration during pregnancy and maternal and neonatal complications such as, preeclampsia, high blood pressure, or small for gestational age infant [3]. Likewise, some interventional and observational studies have shown that vitamin D deficiency or low intake of vitamin D during pregnancy may be associated with an increased risk of developing abnormal glucose tolerance during pregnancy [3,9,10]. However, whether vitamin D during pregnancy also influences the risk of diabetes post GDM is less well studied. Summarizing the current evidence with regard to the potential role of vitamin D on markers of diabetes and diabetes development post GDM is important for disease prevention and drafting public health guidelines. 

Thus, the aim of this systematic literature review was to summarize the current available literature and assess the quality of evidence regarding the associations between vitamin D concentration or supplementation during and after pregnancy and risk of subsequent diabetes development and markers of diabetes development among women with a diagnosis of GDM.

## 2. Method

This systematic review adheres to the Preferred Reporting Items for Systematic Review and Meta-analysis (PRISMA) [11,12]. The international prospective register for systematic reviews, PROSPERO, accepted the protocol for this review on 24 July 2018, registration number: CRD42018102609.

### 2.1. Search Strategy

Databases searched were Medline via PubMed, Cochrane Central Register of Controlled Trials (CENTRAL, Embase, Web of Science (WoS), and Google Scholar. The search was conducted on 12 June 2019. The keywords for the search were applied as Medical Subject Heading (MeSH) terms and as a free search. Considering the format differences for each database, the combination of keywords was the same in all of them: (“25-Hydroxyvitamin D” OR “Calcitriol” OR “Ergocalciferols” OR “Cholecalciferol” OR “Calcifediol” OR “Vitamin D Deficiency”) AND (“Diabetes, Gestational” OR “Gestational Diabetes” OR “Pregnancy in Diabetics”) AND (“Diabetes, Type II” OR “Blood Glucose” OR “Insulin Resistance” OR “Insulin-Secreting Cells”).

Google Scholar was the only database, which presented some difficulties with this combination of terms. The keywords were too many; the search specialist automatically eliminated the two last terms. Furthermore, the results were too many in comparison with the other databases, an indicator of the inclusion of too many irrelevant articles. For these reasons, after trying different combinations, the best results were achieved keeping the three main terms required for the search: (“25-Hydroxyvitamin D” AND “Gestational Diabetes” AND “Diabetes, Type II”).

### 2.2. Study Selection

The studies generated from the defined search strategy were imported from Endnote into Covidence and duplicates were removed. After removing the duplicates, two independent reviewers (C.V.V., M.N.H.) evaluated the titles and abstracts for the articles following the pre-specified criteria. Full texts identified in the first step were screened independently by the same two reviewers. Finally, any disparities were solved at a meeting, including the two reviewers and a third reviewer (A.K.). Reference lists of selected articles were also searched.

The considered studies had to fulfil the following criteria:-Population: This review considered studies where the population involved were pregnant women diagnosed with GDM, ≥18 years of age, and without a previous diagnosis of T2DM. There was no restriction regarding the methods to diagnose GDM.-Intervention: Vitamin D intervention or exposure was the target in this review. The considered sources of maternal vitamin D were vitamin D in food or supplements; and, maternal vitamin D concentration from blood or serum samples during and after pregnancy among women with a GDM diagnosis during their pregnancy. The included measurements of vitamin D in this review were dietary intake of D_2_-ergocalciferol and D_3_-cholecalciferol and blood/serum concentration of 25(OH)D_2_, 25(OH)D_3_, or 1.25(OH)D.-Comparator: Relevant comparisons to include in this review were against placebo in studies with vitamin D supplementation; high versus low doses of vitamin D supplementation; higher versus lower 25(OH)D concentrations in blood or serum; dose–response blood or serum concentrations; and no-treatment control group.-Outcome: The diagnosis of T2DM was the primary outcome. Diabetes markers such as (but not limited to) blood glucose, insulin resistance, insulin sensitivity, impaired beta-cell function, and glycated hemoglobin were the secondary outcomes. There were no restrictions regarding measurements methods or units used.-Study design: Observational prospective comparative cohort studies, controlled (non-randomized) clinical trials (CCTs), and randomized clinical trials (RCTs) were considered for this review. Cross-sectional and case–control studies were initially to be excluded, however, due to the paucity of available studies and to provide a more holistic overview of the current literature on the topic, results from these study designs were included. Case series and case reports were excluded. There were no restrictions based on length of follow-up, and animal studies were not included.

### 2.3. Data Extraction

Data extraction was conducted independently by two reviewers (C.V.V., M.N.H.) using a predefined template in Covidence. Additionally, the following information on the descriptive and quantitative characteristics of studies was extracted: (i) characteristics of the study: authorship, year, country, setting, sample size, design, methods, duration of follow-up, source of funding, conflict of interest; (ii) characteristics of the population: age, ethnicity, co-interventions, information regarding respondent bias, or representativeness of included population; (iii) details about the exposure or intervention (e.g., vitamin D supplementation doses); (iv) details about comparator group (e.g., placebo); (v) outcomes: diagnosis and markers of T2DM such as (but not limited to) blood glucose, insulin resistance, insulin sensitivity, impaired beta-cell function, glycated hemoglobin, metabolic syndrome; adjusted and unadjusted effect estimates; and (vi) confounding factors such as body mass index (BMI), energy intake, physical activity, or level of education.

### 2.4. Quality Assessment

The quality of included studies was assessed by two reviewers (R.D., M.N.H.) based on the criterion provided by the Cochrane Collaboration’s tool [13] and the Risk Of Bias In Non-randomized Studies - of Interventions (ROBINS-I) [14] tools for assessing risk of bias of RCTs and prospective observational studies, respectively.

The Cochrane Collaboration’s tool provides seven quality domains [15]. Each domain is classified into three levels of risk of bias (low, high, or unclear) based on specific criteria. The seven domains are as follows: sequence generation; allocation concealment; blinding of participants and personnel; blinding of outcome assessment; incomplete outcome data; selective outcome reporting, and other sources of bias.

The ROBINS-I tool also presents seven quality domains to assess risk of bias: confounding (age, ethnicity, BMI, energy intake, physical activity, level of education), selection of participants into the study, classification of the interventions, deviations from intended interventions, missing data, measurement of outcomes, and selection of the reported results [14]. The first two domains address issues at baseline, the third one during the intervention, and the last four after the intervention. First, each category is evaluated through a sequence of signaling questions. Within each domain, the conclusion on whether the individual studies were rated with a low, moderate, serious, or critical risk of bias was reached. Finally, across the domains, the overall risk of bias was rated based on the classification that indicated the highest risk of bias.

We assessed the certainty in the evidence using The Grading of Recommendations Assessment, Development and Evaluation (GRADE) [16], which was categorized as very low, low, moderate, and high and is an indication of the robustness in the interpretations of the results. While observational studies started at a low certainty level, RCTs started at a high certainty level. Both types of study designs were then assessed for possible downgrading, based on the following domains: overall risk of bias, inconsistency, indirectness, imprecision, and publication bias.

### 2.5. Meta-Analysis

A meta-analysis was planned, however, due to the paucity of available studies and the heterogeneity of the included outcomes, it was deemed inapplicable.

## 3. Results

### 3.1. Literature Search

From the literature search, there were 58 generated records for PubMed, 62 for WoS, 66 for Embase, 19 for CENTRAL, and 743 for Google Scholar. However, since Google Scholar presented some difficulties in the importation phase and the downloading phase, only 255 could be imported.

A total of 104 duplicates were removed, and thus 356 studies were screened by title and abstract. After that, 288 studies did not meet the inclusion criteria and were excluded. Finally, 68 studies were full text screened, but only six met the inclusion criteria. Thus, a total of six studies were included in the review [17,18,19,20,21,22]. A flowchart of included studies is presented in Figure 1, and a list of excluded studies after full text screening including reasons for their exclusion is presented in Appendix A.

### 3.2. Description of the Studies

The six studies that met the criteria to be included in the present review were all conducted between 2012 and 2017 [17,18,19,20,21,22]. The characteristics of the six studies are summarized in Table 1. Two studies were conducted in Iran and the other four in Canada, Sweden, Hungary, and Malaysia, respectively. The total number of participants was 1169 with a mean age of 33.3 years (Table 1). In one of the six included studies [20], the maternal age required to participate in the research was ≥16 years. However, the mean age was 32 ± 5.5 years in the intervention group and 32.4 ± 4.7 years in the control group [20], which indicates a low presence of women younger than 18 years. For that reason, the decision was made not to exclude this study from the review. Out of the six studies, three were RCTs with two intervention arms, while one was a prospective study [22], one a cross-sectional [18] observational study, both including three comparison groups, and the final study was a nested case–control study [19]. All studies included pregnant women with a GDM diagnosis, vitamin D supplementation or concentration during or after pregnancy, and biomarkers associated with the risk of T2DM development. The length of follow-ups ranged from 6 weeks to less than 4 years. The retention rate was high for the RCTs, ranging from 84.6% to 100% (Table 1). Measurements of maternal BMI at baseline across the six studies are presented in Appendix A.

### 3.3. Description of Vitamin D Interventions or Exposure and Comparator Groups

The three RCTs comprised women with GDM. In one of the RCTs, the intervention consisted of one vitamin D (25(OH)D_3_) intramuscular injection of 300.000 IU 3–10 days postpartum (baseline), comparing with a no-treatment control group [17]. In another RCT, the intervention group took vitamin D (25(OH)D_3_) supplementation of 700.000 IU during pregnancy. Participants within a gestational age range of 12 and 27 weeks were instructed to take 200.000 IU of vitamin D_3_ the first two days, and an additional 50.000 IU per week until reaching 700.000 IU, and women in their 28th week of pregnancy or later (max 32) were instructed to take 100.000 IU weekly. Comparisons were made with a no-treatment control group. Women with sufficient basal serum vitamin D were excluded from the trial [20]. Finally, in the third RCT, participants in the intervention group took capsules of 25(OH)D_3_ of 4000 IU per day for 6 months, which was stated to start 6–48 months after pregnancy. The comparator group took placebo capsules during the same period [21] (Table 1).

In the three observational studies, vitamin D status was assessed with the measurement of (serum) 25(OH)D concentration at 1–2 years postpartum in the study by Shaat et al. [18], at 3 months postpartum in the study by Kramer et al. [22], and at 3.2 years postpartum in the study by Tänczer et al. [19]. In the studies by Shaat et al. [18] and Kramer et al. [22], between-group comparisons of 25(OH)D concentrations based on the following cutoffs from the Endocrine Society guidelines: deficiency (<50 nmol/L), insufficiency 50–74 nmol/L, and sufficiency ≥75 nmol/L were made [23]; whereas 25(OH)D was used as a continuous variable in the study by Tänczer et al. [19] (Table 1).

Therefore, of the six included studies, only one assessed the effect of vitamin D (through supplementation) during pregnancy in relation to markers of T2DM after GDM [20], while the remaining five studies included 25(OH)D measurements or supplementation after pregnancy.

### 3.4. Vitamin D Concentrations

Two of the three RCTs reported median concentration of 25(OH)D while the third one [20] reported mean 25(OH)D concentration. In the intervention groups, median (25th, 75 pct) pre- and post-intervention 25(OH)D concentrations were higher in the study from Yeow et al. [21] compared to that from Mozaffari-Khosravi et al. [17] (35.6 (25.60, 43.95); 92.4 (79.00, 102.34) (*p* = 0.003) and 24.25 (17.05, 28.2); 62.10 (55.47, 71.70) (*p* < 0.001), respectively). This was also true in the control groups. Mean 25(OH)D concentration was lower in the intervention group compared to that in the control group at baseline, and the opposite was observed at follow-up in the study by Valizadeh et al. [20] (Table 2).

The mean 25(OH)D (nmol/L) concentrations were 35.7 ± 10.2, 64.4 ± 7.4, and 91.2 ± 12.5 (*p* < 0.001) in the deficient, insufficient, and sufficient groups, respectively, in the study by Kramer et al. [22] and 32.9 ± 11.2, 60.8 ± 7.1, 88.1 ± 11.2 (*p* < 0.001), respectively, in the study by Shaat et al. [18]. In the nested case–control study by Tänczer et al. [19] mean 25(OH)D concentration levels were similar among cases and controls (27.2 ± 13.1; 26.9 ± 9.8 ng/mL equivalent to: 68 ± 32.75; 67.25 ± 24 nmol/L, *p* = 0.888) (Table 2). 

### 3.5. Synthesis of Results

#### 3.5.1. Insulin Sensitivity and Resistance

All six studies [17,18,19,20,21,22] included measurements of insulin sensitivity and/or resistance among women with previous GDM. Mixed findings were reported between vitamin D supplementation or concentration and different makers of insulin sensitivity and resistance (Table 3). 

#### 3.5.2. Beta-Cell Function

All six studies [17,18,19,20,21,22] included measurements of β-cell function among women with previous GDM. Most markers of β-cell function were not associated with vitamin D supplementation or concentration (Table 3).

#### 3.5.3. Glucose Measurements

Four [17,20,21,22] out of six studies included glucose measurements among women with previous GDM. The three RCTs reported no associations between vitamin D supplementation and change in different glucose measurements; whereas, Kramer et al. found inverse associations between 25(OH)D concentration and indicators of glucose metabolism in the vitamin D-deficient group compared to the sufficient group and mixed findings among the insufficient group (Table 3).

#### 3.5.4. Glycated Hemoglobin

Glycated hemoglobin (HbA1c) among women with previous GDM was assessed in the three RCTs [17,20,21] and none of them showed an association between vitamin D supplementation and HbA1c (Table 3).

#### 3.5.5. Diabetes

Development of diabetes after GDM was investigated cross-sectionally in two studies [18,20], neither of which found an association with vitamin D concentration after pregnancy and diabetes (Table 3).

When considering results from RCTs only [17,20,21], vitamin D supplementation (during and after pregnancy) was not associated with markers of diabetes [20] or change in diabetes’ markers [17,21] or diabetes development after GDM, apart for some markers of insulin sensitivity and insulin resistance, where statistically significant direct and/or inverse associations were reported in some studies (Appendix A).

### 3.6. Risk of Bias within the Six Included Studies

Appendix A provides a risk-of-bias summary of included observational studies and RCTs. Most of the domains presented a low risk of bias in all three RCTs. Blinding of participants, personnel, and outcome assessors were the most complicated domains, however since the reporting of the outcomes of interest in the present review is not likely to be influenced by the knowledge of group allocation, these were rated low. In general, the study by Mozaffari et al. [17] was insufficiently described according to the Cochrane risk-of-bias domains.

For the observational studies, the most problematic domain was bias due to confounding, where one study [22] was assessed as serious and the two others as critical [18,19]. The rest of the categories were evaluated as low-to-moderate risk of bias, except for bias due to departures from intended interventions, where there was not enough information to make a judgment. The only exception was the study by Tänczer et al., where bias due to missing data was rated as critical risk of bias. Thus, considering the overall quality assessment, one article was evaluated as serious risk of bias [22] and the two others as critical risk of bias [18,19].

## 4. Discussion

In this systematic review, quality assessment and summary of results from the six published studies that examined the associations between vitamin D concentration or supplementation during and after pregnancy and risk of subsequent diabetes development and markers of diabetes development among women with a diagnosis of GDM were carried out. The methodological quality of published studies using the Cochrane Collaboration’s [15] and ROBINS-I [14] tools for assessing risk of bias of RCTs and observational studies, respectively, was performed, and a suboptimal quality for observational studies was found; whereas randomized interventional studies were evaluated as having a low risk of bias.

Findings from the six included studies on the associations between vitamin D supplementation or concentration and markers of diabetes development or diabetes development post GDM were inconsistent across studies. However, when considering randomized interventional studies only, the findings more strongly suggested that vitamin D supplementation during and after pregnancy did not have an influence on markers of diabetes development or diabetes development post GDM.

### 4.1. Context with Previous Published Results

To the best of our knowledge, no previous systematic review has provided a synthesis of the literature regarding the association between vitamin D and the risk of subsequent diabetes development post GDM. Previous reviews and meta-analyses have focused on the role of vitamin D concentration or supplementation on the development of GDM [9,24,25]. Evidence from one of these meta-analyses suggested that pregnancies with low blood vitamin D had a higher risk of GDM (odds ratio 1.85) and that vitamin D supplementation during pregnancy significantly reduced fasting plasma glucose (FPG), fasting insulin levels, and HOMA-IR [24]. On the other hand, results from the Diabetes and Pregnancy Vitamin D and Lifestyle Intervention for Gestational Diabetes Mellitus Prevention (DALI) European multicenter randomized controlled trial found only a minor beneficial effect of vitamin D supplementation on glucose metabolism [26]. These discrepancies might be due to differences in baseline vitamin D concentrations. Indeed, in the review by Zhang et al. [24], a positive effect of vitamin D concentration was seen among RCTs where the participants were vitamin D deficient and vitamin D doses in the intervention arm were high [24,27]. In contrast, the DALI study population had higher rate of vitamin D sufficiency at baseline and showed a minimal effect of vitamin D supplementation on markers of GDM development [26]. In the current review, participants from the three included RCTs had 25(OH)D concentration <50 nmol/L at baseline, with participants in the intervention arms reaching vitamin D insufficiency in one RCT [17] and sufficiency in the two others [20,21], while the control groups remained deficient. The three RCTs included high doses of vitamin D supplementation in the intervention arms of between 300.000 IU at once to 700.000 IU and 720.000 IU over six months. In the three included observational studies [18,19,22], most participants were vitamin D deficient or insufficient. Hence, this suggests that the general lack of associations/effects found in this review might not be ascribed to high baseline vitamin D concentrations or low doses of vitamin D supplementation but rather may depend on the heterogeneity of the studies. In this regard, also timing (during and after pregnancy) of 25(OH)D measurements/supplementation, methods of (serum) 25(OH)D assay measurements as well as modes (oral or intramuscular) of vitamin D supplementations were different across the six included studies, as well as between RCTs, which makes a direct comparison difficult as the effects/biological responses might be disparate. Likewise, a broad diversity of diabetes markers and measurement methods were used in the included studies, which may have further contributed to the low ability to draw any firm conclusions in this review. 

Among non-pregnant diabetic individuals, vitamin D supplementation has in previous studies shown beneficial effects in terms of insulin resistance [28,29], fasting glucose [29], and HbA1C [30]. In a systematic review of meta-analyses and randomized trials of the effect of vitamin D supplementation on non-skeletal disorders, Autier et al. [31] concluded that, regardless of the individuals’ 25(OH)D status at baseline, vitamin D supplementation did not show any benefit on glucose metabolism biomarkers or on diabetes progression. Hence, despite the many studies, the role of vitamin D in the development of diabetes outside and during pregnancy remains unclear; and regarding the role of vitamin D on markers of diabetes development and diabetes development post GDM, current evidence, based on a few studies, also does not support associations.

This research field is sparsely investigated, which is reflected in the low number of studies eligible for inclusion in the present review. Further, included studies generally had low quality ratings. The observational studies included were of low to very low quality evidence. Thus, the conclusions hold a high degree of uncertainty from the observational data. Particularly the methodological challenges were clear from the results of our risk of bias assessment, especially in relation to confounding. When evaluating the risk of bias due to confounding using ROBINS-I, the assessment is based on a pre-specified listing of the confounding domains that are relevant to all or most of the studies eligible for the review. Removing studies with critical risk of bias (as recommended in ROBINS-I was deliberately disregarded in this systematic review because of the few studies identified. The randomized interventional studies were of moderate quality of evidence. Although these studies had limited methodological challenges, the sample sizes were small, and the follow-up periods may not have been of sufficient length, as different adverse metabolic states may not have had time to develop as quickly as under 6 months of time. As such, based on the current evidence, it is not possible to conclude whether vitamin D plays a role in later risk of diabetes among patients with GDM. Future observational studies should focus on their methodological quality, particularly concentrating on controlling for potential confounding factors as well as including pertinent outcome measurements and improving the selection of reported results. Future RCTs would benefit from including larger sample sizes and longer follow-up periods. 

### 4.2. Strengths and Weaknesses of This Systematic Literature Review

One of the strengths of this review lies in its systematic approach and quality assessment of the included studies. Following the PRISMA guidelines, the review protocol was registered onto Prospero prior to the search; furthermore, the study selection and quality assessment were performed independently by two researchers. A comprehensive literature search was performed; however, we did not assess the risk of publication bias, and as we only selected published articles written in English, issues regarding publication bias should be kept in mind.

Conducting a meta-analysis was deemed difficult because of the paucity and heterogeneity of the included studies, therefore vote counting was used as a synthesis method. Hence, it was only possible to assess whether there was any evidence of an effect or association rather than to evaluate the average intervention/exposition effect [32]. 

## 5. Conclusions

In conclusion, this systematic review highlights inconsistent findings on the associations between vitamin D supplementation or concentration during and after pregnancy and markers of diabetes development or diabetes development post GDM; and although results from randomized interventional studies more strongly suggested no associations, the conclusion holds a high degree of uncertainty.

## Figures and Tables

**Figure 1 nutrients-12-01733-f001:**
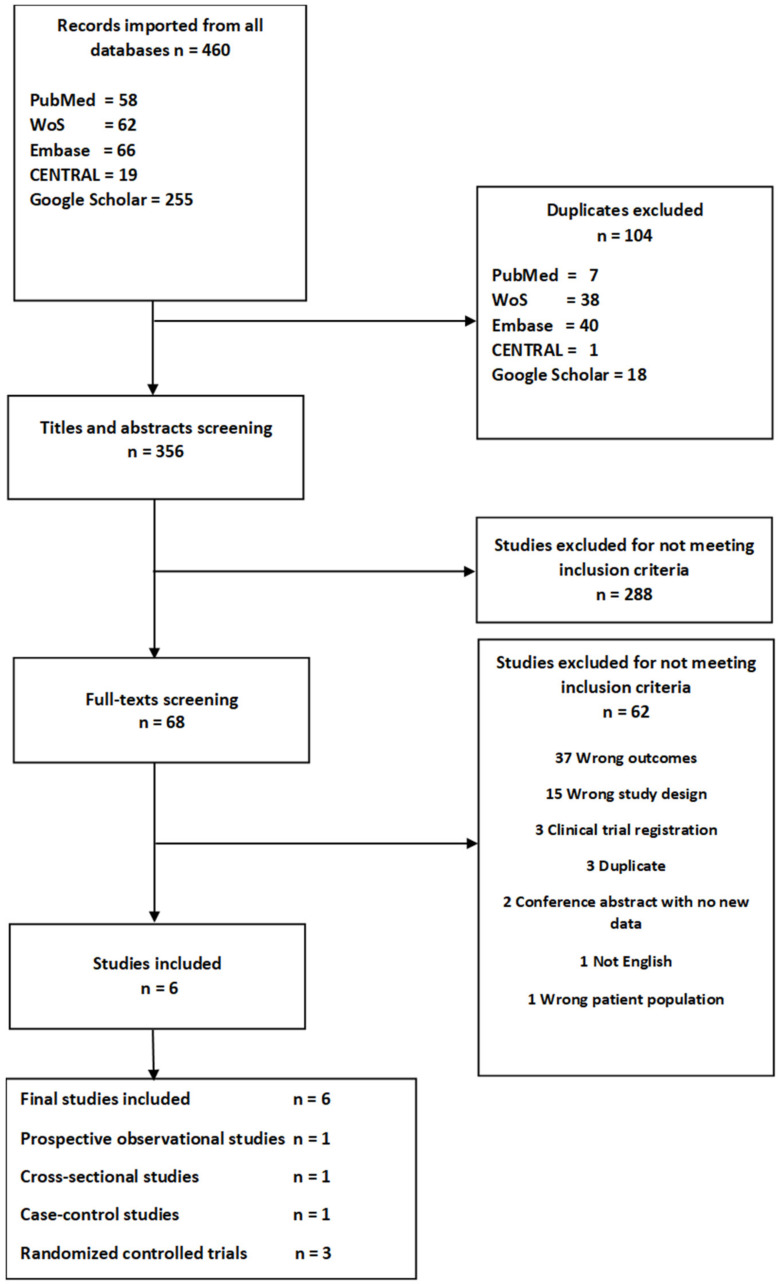
Flowchart.

**Table 1 nutrients-12-01733-t001:** Characteristics of the six included studies on vitamin D in gestational diabetes and markers of type 2 diabetes development.

Study’s First Author, Year, Country	Study Design, N, Follow-Up, Retention	Age (Mean ± SD) (Median 25th, 75th Pct)	Ethnicity	Intervention/Exposure	Comparison/Control Group	Reported Outcomes	Results	Conflict of Interest
[22] Kramer et al., 2014, Canada	Prospective cohort study, 494, 3–12 months	34.8 ± 4.3	60.2% Caucasian 39.8% Other	25(OH)D serum concentration measured after pregnancy by competitive electro chemiluminescent immunoassay Classified in three groups: (a) Deficiency (25(OH)D < 50 nmol/L) (b) Insufficiency (25(OH)D ≥ 50 < 75 nmol/L) (c) Sufficiency (25(OH)D ≥ 75 nmol/L	Comparisons between 25(OH)D status groups with vitamin D sufficiency as reference	*Insulin sensitivity*		No potential conflicts of interest relevant to this article were reported
						Matsuda index	^1^ Deficient −0.099, *p* = 0.08 ^1^ Insufficient −0.013, *p* = 0.79	
						*β-cell function*		
						ISSI-2	^1^Deficient −0.062, *p* = 0.16 ^1^ Insufficient −0.111, *p* = 0.006	
						*Glucose measurements*		
						FBG on OGTT (nmol/L)	^1^ Deficient 0.026, *p* = 0.008 ^1^ Insufficient 0.011, *p* = 0.20	
						2 hour 75 g OGTT (mg/dL)	^1^ Deficient 0.070, *p* = 0.01 ^1^ Insufficient 0.051, *p* = 0.04	
[19] Tänczer et al., 2017, Hungary	Nested case–control, 132, 3.2 (±0.6) years	Cases 34.8 ± 4.4 Controls 33.8 ± 3.6	100% Caucasian	* 25(OH)D concentration measured after pregnancy by chemiluminescent immunoassay (CLIA) method, used continuously (ng/mL)	Comparisons between women with GDM and a randomly selected control group of women with normal glucose tolerance during pregnancy	*Insulin sensitivity*		No potential conflicts of interest relevant to this article were reported
						HOMA2-S	^2^ β 0.017 (95%CI 0.003, 0.031), *p* = 0.02	
						*β-cell function*		
						HOMA2-B	^2^ β –0.009 (95%CI 0.001, 0.018), *p* = 0.085	
[18] Shaat et al., 2017, Sweden	Cross-sectional, 376, 12–24 months	34.3 ± 4.8	79% Caucasian 10% Asian 9% Arab 2% Other	25(OH)D serum concentration measured by liquid chromatography mass spectrophotometry after pregnancy. Classified in three groups: (a) Deficiency (25(OH)D < 50 nmol/L) (b) Insufficiency (25(OH)D ≥ 50 < 75 nmol/L) (c) Sufficiency (25(OH)D ≥ 75 nmol/L)	Comparisons between 25(OH)D status groups	Insulin resistance		The authors have stated explicitly that there are no conflicts of interest in connection with this article
						HOMA-IR (median, IQR)	^3^ Deficient: 1.8 (1.1–2.7) Insufficient: 1.6 (1.0–2.3) Sufficient: 1.1 (0.8–1.8) *p* = 0.001	
						β-cell function (median, IQR)		
						Insulinogenic index	^3^ Deficient: 12.1 (7.7–20.4) Insufficient: 12.4 (8.5–18.9) Sufficient: 11.1 (8.4–18.2) *p* = 0.730	
						^£^ Disposition index (I/G30)/HOMA-IR	^3^ Deficient: 8.1 (4.9–12.3) Insufficient: 8.5 (5.4–14.0) Sufficient: 10.1 (5.4–14.6) *p* = 0.035	
						^¥^ T2DM after GDM (OR, 95% CI)	^4^ 1.0 (1.0, 1.1), *p* = 0.130	
[17] Mozaffari-Khosravi et al., 2012, Iran	RCT, 45, 3 months, 45/45 (100%)	Intervention group 30.7 ± 6.2 Control group 29.5 ± 4.0	Not mentioned	One intramuscular injection of 300.000 IU 3-10 days after pregnancy. 25(OH)D concentration measured by immunoassay method (NycoCard; Nyco Corporation, Oslo, Norway)	No-treatment control group	Insulin sensitivity		The authors have stated that they had nothing to declare in connection with this article
						QUICKI (mean ± SD)	^5^ Intervention: 0.38 ± 0.02 Control: 0.36 ± 0.02, *p* = 0.006	
						HOMA-S (mean ± SD)	^5^ Intervention: 169.68 ± 53.48 Control: 122.84 ± 41.15, *p* = 0.002	
						Insulin resistance		
						HOMA-IR (25th, 50th, 75th pct)	^6^ Intervention: 0.4, 0.5, 0.8 Control: 0.7, 0.9, 1.0, *p* = 0.004	
						*β-cell function* HOMA-B (mean ± SD)	^5^ Intervention: 69.97 ± 28.13 Control: 77.68 ± 34.44, *p* = 0.40	
						Glucose measurements		
						FBG (mg/dL) (mean ± SD)	^5^ Intervention: 92.9 ± 10.6 Control: 104.7 ± 33.5, *p* = 0.11	
						2 h 75 g OGTT (mg/dL) (mean ± SD)	^5^ Intervention: 123 ± 69.04 Control: 117 ± 56.3, *p* = 0.8	
						Glycated hemoglobin		
						HbA1c (nmol/mol) (mean ± SD)	^5^ Intervention: 37 ± 13 Control: 34 ± 6, *p* = 0.22	
						HbA1c (%) (mean ± SD)	^5^ Intervention: 5.58 ± 1.2 Control: 5.21 ± 0.52, *p* = 0.2	
[20] Valizadeh et al., 2016, Iran	RCT, 96, 6–12 weeks, 84/96 (87.5%)	Intervention group 32.0 ± 5.5 Control group 32.4 ± 4.7	Not mentioned	Oral vitamin D supplementation of 700.000 IU from 12 gestational weeks until delivery. 25(OH)D concentration measured by ELISA (enzyme-linked immunosorbent assay) method (Immunodiagnostic Systems Ltd., Tyne and Wear, UK)	No-treatment control group	Insulin resistance		This study was supported by a studentship fund from Zanjan University of Medical Sciences. Farir-Teb Company supported this study by providing glucometers (Glucocard 0-1) for the parents
						HOMA-IR (mean ± SD)	^7^ Intervention: 2.0 ± 1.3 Control: 1.8 ± 1.9, *p* = 0.58	
						β-cell function		
						Serum insulin level (µu/mL) (mean±SD)	^7^ Intervention: 8.7 ± 4.4 Control: 8.8 ± 9.7, *p* = 0.99	
						Glucose measurements		
						FPG (mg/dL) (mean±SD)	^7^ Intervention: 94 ± 16 Control: 89 ± 13, *p* = 0.12	
						2 h 75 g OGTT (mg/dL) (mean ± SD)	^7^ Intervention: 115 ± 48 Control: 110 ± 36, *p* = 0.56	
						Glycated hemoglobin		
						HbA1c (%) (mean ± SD)	^7^ Intervention: 5.6 ± 0.5 Control: 5.5± 0.5, *p* = 0.24	
						HbA1c (nmol/mol) (mean ± SD)	^7^ Intervention: 38 ± 5 Control: 37 ± 5, *p* = 0.24	
						^ɕ^ Dysglycemia (OR, 95% CI)	^8^ 1.02 (0.98, 1.06), *p* = 0.337	
[21] Yeow et al., 2015, Malaysia	RCT, 26, 6 months, 22/26 (84.6%)	Intervention group 36 (32, 38) Control group 35 (30, 40)	100% Asian	Oral Vitamin D supplementation (capsules) of 4000 IU per day for 6 months, 6–48 months after pregnancy. 25(OH)D concentration measured the Elecsys Vitamin D Total assay (Roche Diagnostics GmbH, Sandhofer Strasse 116, D-68305 Mannheim, Germany)	Placebo capsules for 6 months, 6–48 months post-partum	Insulin sensitivity		The vitamin D and placebo capsules were supplied by Blackmore Ltd. without charge. This does not alter the author’s adherence to PLOS ONE policies on sharing data and materials. The authors declare no competing interest between all study investigators and Blackmore Ltd. in terms of employment, consultancy, and patents of the product or its development
						QUICKI (Δ Median (25th, 75th pct))	^6^ Intervention: −0.01 (−0.029, 0.01) Control: 0.01 (−0.005, 0.021), *p* = 0.047	
						OGIS (mL/min/m^2^) (Δ Median (25th, 75th pct))	^6^ Intervention: −10 (−47.0, 55.0) Control: −1 (−82.0, 42.0), *p* = 1.00	
						BIGTT-S (10–5 × (min × pmol/L)−1) (Δ Median (25th, 75th pct))	^6^ Intervention: −0.6 (−1.15, 0.60) Control: −0.3 (−1.29, 1.04), *p* = 0.699	
						Insulin resistance		
						Fasting insulin (pmol/L) (Δ Median (25th, 75th pct))	^6^ Intervention: 15.6 (−13.80, 51.00) Control: −14.4 (−46.20, 0), *p* = 0.034	
						Fasting C-peptide (ng/mL) (Δ Median (25th, 75th pct))	^6^ Intervention: 0.4 (0.03, 0.54) Control: 0.3 (−0.09, 0.48), *p* = 0.365	
						*Glycated hemoglobin* HbA1c (nmol/mol) (Δ Median (25th, 75th pct))	^6^ Intervention: −2 (−3, −1) Control: −2 (−4, 0), *p* = 0.847	
						Glucose measurements		
						FPG (Δ Median (25th, 75th pct))	^6^ Intervention: 0 (−0.20, 0) Control: 0.1 (−0.20, 0.50), *p* = 0.270	
						30 min 75 g OGTT (mmol/L) (Δ Median (25th, 75th pct))	^6^ Intervention: −0.4 (−2.40, 1.70) Control: −0.1 (−1.93, 0.48), *p* = 1.00	
						2 h 75 g OGTT (mmol/L) (Δ Median (25th, 75th pct))	^6^ Intervention: −2.6 (−1.50, −0.40) Control: 0.4 (−1.20, 1.00), *p* = 0.061	
						AUCglucose (mmol/L) (Δ Median (25th, 75th pct))	^6^ Intervention: −28.5 (−199.50, 70.5) Control: −56.6 (−130.5, 54.0), *p* = 1.00	
						β-cell function		
						AUCinsulin (pmol/L) (Δ Median (25th, 75th pct))	^6^ Intervention: 17,376 (−8574, 41,514) Control: 3894 (−10,242, 17,524) *p* = 0.365	
						AUCcp (ng/mL) (Δ Median (25th, 75th pct))	^6^ Intervention: 157 (79.0, 210.0) Control: 134 (86.0, 269.0), *p* = 1.00	
						IGI_60_ (pmol/mmol) (Δ Median (25th, 75th pct))	^6^ Intervention: 31.8 (−30.05, 297.33) Control: 82.66 (−2.00, 203.87), *p* = 0.863	
						BIGTT-AIR (min × pmol/L) (Δ Median (25th, 75th pct))	^6^ Intervention: 1241.2 (−299.48, 2260.43) Control: −144.8 (−1893.53, 916.62), *p* = 0.133	
						Disposition index (OGIS * ratio of total AUCinsulin over AUCglucose) (Δ Median (25th, 75th pct))	^6^ Intervention: 7.7 × 10^3^ (2.94 × 10^3^, 17.52 × 10^3^) Control: 4.5 × 10^3^ (−3.88 × 10^3^, 10.70 × 10^3^), *p* = 0.171	

HOMA-IR: homeostasis model assessment of insulin resistance; QUICKI: quantitative insulin sensitivity check index; OGTT: oral glucose tolerance test; FBG: fasting blood glucose FPG: fasting plasma glucose; HOMA2-S: homeostasis model assessment insulin sensitivity; HOMA2-B: homeostasis model assessment beta-cell function; ISSI-2: insulin secretion-sensitivity index-2; HbA1c: hemoglobin A1c; IQR: interquartile range; OR: odds ratio; SD: standard deviation; Δ Median (25th, 75th pct): median between-group change between baseline and follow-up (endpoint–baseline); pct: percentiles; OGIS: oral glucose insulin sensitivity index; AUC: area under the curve; AUCcp: area under the curve of C-peptide; IGI_60_: insulinogenic index calculated at 60 min; BIGTT: pancreatic beta-cell function, insulin sensitivity, and glucose tolerance test; BIGTT-S: BIGTT with insulin sensitivity; BIGTT-AIR: BIGTT with acute insulin response. * 25(OH)D samples (blood, plasma, serum) unclear. ^£^ (I/G30) is the ratio of the incremental insulin to glucose during the first 30 min of the OGTT, i.e., (insulin_30 min_ – insulin_0 min_)/(glucose_30 min_ – glucose_0min_). ^¥^ Based on the WHO 1999 criteria: fasting 2 h 75 g OGTT: ≥140 mg/dL (7.8 mmol/L). ^ɕ^ Dysglycemia was defined as the development impaired fasting glucose (IFG) or impaired glucose tolerance (IGT) or type 2 diabetes in subjects as measured by the postpartum tests. IFG was defined by FPG levels of 100 to 125 mg/dL, IGT by 2-hPLG levels of 140–199 mg/dL, and type 2 diabetes by FPG levels ≥126 or 2-PLG levels ≥200 mg/dL. ^1^ Multiple linear regression analyses with estimates and *p*-value, log-transformed outcomes, adjusted for age, ethnicity, family history of T2DM, previous GDM, BMI, fasting glucose at 3 months, duration of breastfeeding, physical activity, and season (model 3), with 25(OH)D sufficiency as reference group. ^2^ Multiple linear regression adjusted for age and waist circumference with log-transformed HOMA2-S and HOMA2-B. ^3^ ANOVA. ^4^ Multivariable model adjusted for BMI, non-European origin, HOMA-IR, and insulinogenic index. ^5^ Student t-test. ^6^ Mann–Whitney U-test. ^7^ Independent samples *t*-test. ^8^ Logistic regression.

**Table 2 nutrients-12-01733-t002:** Baseline and post-intervention concentrations of vitamin D (25(OH)D (nmol/L)) in the six included studies.

Study (Design)		*N* (%)	Median (25th, 75th Percentiles) or Mean ± SD	*p*-Value Difference within Group	*N*	Median (25th, 75th Percentiles) or Mean ± SD	*p*-Value Difference within Group	Between Group *p*-Value
[22] Kramer et al.	Deficiency	161 (33)	35.7 ± 10.2			-	-	-
(cohort)	Insufficiency	178 (36)	64.4 ± 7.4			-	-	-
	Sufficiency	155 (31)	91.2 ± 12.5	-		-	-	<0.001
[18] Shaat et al.	Deficiency	198 (53)	32.9 ± 11.2			-	-	-
(cross-sectional)	Insufficiency	125 (33)	60.8 ± 7.1			-	-	-
	Sufficiency	53 (13)	88.1 ± 11.2	-		-	-	<0.001
			Cases			Controls		
[19] * Tänczer et al.		87	68 ± 32.75		45	67.25 ± 24		0.888
(case-control)								
			Intervention group			Control group		
[21] Yeow et al.	Baseline	13	35.6 (25.60, 43.95)	0.003	13	35.1 (21.63, 40.75)	0.859	
(RCT)	End		92.4 (79.00, 102.34)			28.5 (20.87, 42.43)		<0.001
[17] Mozaffari-Khosravi et al.	Baseline	24	24.25 (17.05, 28.2)	<0.001	21	25.3 (20.0, 32.35)	0.02	
(RCT)	End		62.10 (55.47, 71.70)			24.1 (21.70, 48.60)		<0.001
[20] * Valizadeh et al.	Baseline	42	36.5 ± 15.75		42	44.25 ± 15.25		0.04
(RCT)	End		81 ± 36			48.25 ± 24		<0.001

SD: standard deviation. * 25(OH)D concentration in ng/mL was transformed to nmol/L.

**Table 3 nutrients-12-01733-t003:** Synthesis of results with direction of associations by outcome among the six included studies on vitamin D in gestational diabetes and markers of type 2 diabetes development.

Outcomes	Study	Direction of Associations
**Insulin sensitivity**		
Matsuda index	Kramer et al. [22]	* Deficiency: no
		* Insufficiency: no
HOMA2-S	Tänczer et al. [19]	(+)
HOMA-S	Mozaffari-Khosravi et al. [17]	(+)
QUICKI	Mozaffari-Khosravi et al. [17]	(+)
	Yeow et al. [21]	(−)
OGIS	Yeow et al. [21]	No
BIGTT-S	Yeow et al. [21]	No
**Insulin resistance**		
HOMA-IR	Shaat et al. [18]	(−)
	Mozaffari-Khosravi et al. [17]	(−)
	Valizadeh et al. [20]	No
Fasting insulin	Valizadeh et al. [20]	No
	Yeow et al. [21]	(+)
Fasting C-peptide	Yeow et al. [21]	No
**Beta-cell function**		
HOMA2-B	Tänczer et al. [19]	No
HOMA-B	Mozaffari-Khosravi et al. [17]	No
ISSI-2	Kramer et al. [22]	* Deficiency: no
		* Insufficiency: (+)
Insulinogenic index	Shaat et al. [18]	No
Disposition index	Shaat et al. [18]	(+)
	Yeow et al. [21]	No
AUCinsulin (pmol/L)	Yeow et al. [21]	No
AUCcp (ng/mL)	Yeow et al. [21]	No
IGI_60_ (pmol/mmol)	Yeow et al. [21]	No
BIGTT-AIR	Yeow et al. [21]	No
**Glucose measurements**		
FBG	Kramer et al. [22]	* Deficiency: (−)
		* Insufficiency: no
	Mozaffari-Khosravi et al. [17]	No
FPG	Valizadeh et al. [20]	No
	Yeow et al. [21]	No
30 min 75 g OGTT	Yeow et al. [21]	No
2 h 75 g OGTT	Kramer et al. [22]	* Deficiency: (−)
		* Insufficiency: (−)
	Mozaffari-Khosravi et al. [17]	No
	Valizadeh et al. [20]	No
AUCglucose (mmol/L)	Yeow et al. [21]	No
**Glycated hemoglobin**		
HbA1c	Mozaffari-Khosravi et al. [17]	No
	Valizadeh et al. [20]	No
	Yeow et al. [21]	No
% HbA1c	Mozaffari-Khosravi et al. [17]	No
	Valizadeh et al. [20]	No

**Diabetes**	** Shaat et al. [18]	No
	*** Valizadeh et al. [20]	No

No: no statistically significant association; (+) direct/positive association; (−) inverse association. HOMA-IR: homeostasis model assessment of insulin resistance; QUICKI: quantitative insulin sensitivity check index; ISSI-2: insulin secretion-sensitivity index-2; OGTT: oral glucose tolerance test; HOMA(2)-S: homeostasis model assessment insulin resistance; HOMA(2)-B: homeostasis model assessment beta-cell function; FBG: fasting blood glucose; FPG: fasting plasma glucose; HbA1c: hemoglobin A1c; OGIS: oral glucose insulin sensitivity index; AUC: area under the curve; AUCcp: area under the curve of C-peptide; IGI60: insulinogenic index calculated at 60 min; BIGTT: pancreatic beta-cell function, insulin sensitivity, and glucose tolerance test; BIGTT-S: BIGTT with insulin sensitivity; BIGTT-AIR: BIGTT with acute insulin response. * 25(OH)D sufficiency as reference group. ** Based on the WHO 1999 criteria: fasting 2 h 75 g OGTT: ≥140 mg/dL (7.8 mmol/L). *** Dysglycemia was defined as the development impaired fasting glucose (IFG) or impaired glucose tolerance (IGT) or type 2 diabetes in subjects as measured by the postpartum tests. IFG was defined by FPG levels of 100–125 mg/dL, IGT by 2-hPLG levels of 140–199 mg/dL, and type 2 diabetes by FPG levels ≥126 or 2-PLG levels ≥200 mg/dL.

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
