# Peer review of "The Role of Vitamin D in the Development of Diabetes Post Gestational Diabetes Mellitus: A Systematic Literature Review"

_nutrients, 2020, doi:10.3390/nu12061733_

Round 1
Reviewer 1 Report
The present paper deals with a potential interesting topic.
The main limitations are the great heterogeneity among studies, the low number of studied individuals, and the low methodological quality of the revised papers. All these limits were recognized by the authors themselves.
Minor points
GMD instead of GDM in many parts of the manuscript
The review methodology seems correct and the text was clear.
I only suggest to better analyse the limitations of the present systematic review linked to the limitations of the existing studies.
Reviewer 2 Report
The authors present a fine comprehensive review about the topic of vitamin D supplementation in women with GDM, which is under debate within the field. Overall, the authors do a fine job clearly describing their methodology and findings. The introduction covered enough background without being too lengthy.
While they provide in depth details about their selection criteria, it is unfortune and surprising that only 6 studies remain. Therefore, it is important that the authors include an outline of what the ideal cohort study(ies) would be to better address this question of Vitamin D importance/role which is still up for debate and unclear. Perhaps with a flow chart or brief section in the conclusion.
While the authors discuss that BMI was consider in the modeling/analysis could the authors provide the maternal BMI mean and rage for the studies?
In the tables (Tables 1-3), the authors should include ‘vitamin D’ in the title which would help for readability and for future referencing. Suggestions for example, Table 2 …..vitamin D-25(OH)D; Tables 1 and 3…… studies related to vitamin D.
In the final version, Table 3 should not be split up onto separate pages.
Overall manuscript needs to be checked for typos: e.g. line 350 add ‘the’ before individuals’; line 184 add ‘of’ before women; check when referring to dosage IU how the amounts are written and choose one consistent way to write this. (lines 181-192- is written 100000 IU (no comma) and lines 334-335 it is written as 300’000 IU)?
What is the text and the placement after 272 regarding? A footnote? A table? This is confusing.
What is the bold Risk of bias with studies regarding (a table or a section)? This is confusing.
The authors provide good supplemental material.
